# Modeling and Performance Analysis of Satellite Network Moving Target Defense System with Petri Nets

**Leyi Shi** [1,2,*,†], **Shanshan Du** [1,†], **Yifan Miao** [1] and **Songbai Lan** [1]

[1] College of Computer Science and Technology, China University of Petroleum, Qingdao 266580, China; s19070013@s.upc.edu.cn (S.D.); s20070002@s.upc.edu.cn (Y.M.); s20070005@s.upc.edu.cn (S.L.)
[2] Guangxi Key Laboratory of Cryptography and Information Security, Guilin University of Electronic Technology, Guilin 541004, China
[*] Correspondence: shileyi@upc.edu.cn
[†] These authors contributed equally to this work.

**Abstract:** With the development of satellite communication networks and the increase of satellite services, security problems have gradually become some of the most concerning issues. Researchers have made great efforts, including conventional safety methods such as secure transmission, anti-jamming, secure access, and especially the new generation of active defense technology represented by MTD. However, few scholars have theoretically studied the influence of active defense technique on the performance of satellite networks. Formal modeling and performance analysis have not been given sufficient attention. In this paper, we focus on the performance evaluation of satellite network moving target defense system. Firstly, two Stochastic Petri Nets (SPN) models are constructed to analyze the performance of satellite network in traditional and active defense states, respectively. Secondly, the steady-state probability of each marking in SPN models is obtained by using the isomorphism relation between SPN and Markov Chains (MC), and further key performance indicators such as average time delay, throughput, and the utilization of bandwidth are reasoned theoretically. Finally, the proposed two SPN models are simulated based on the PIPE platform. In addition, the effect of parameters on the selected performance indexes is analyzed by varying the values of different parameters. The simulation results prove the correctness of the theoretical reasoning and draw the key factors affecting the performance of satellite network, which can provide an important theoretical basis for the design and performance optimization of the satellite network moving target defense system.

**Keywords:** satellite network; moving target defense; stochastic Petri nets; Markov chain; performance analysis

## 1. Introduction

With the rapid development of aerospace and wireless communication technology as well as the gradual deepening of information construction, the space satellite network is developing at an unprecedented speed. As an important link network in the international communication network, satellite network is widely applied in remote sensing, detection, meteorology, communication, navigation, emergency rescue and other fields, as shown in Figure 1. Especially in the field of communication, which is an important means of information transmission and exchange in human social life, satellite communication, as a supplementary communication method of terrestrial communication, has achieved great success and has become an indispensable part of people's life. Under the dual action of social demand and technology development, satellite communication in the 21st century is climbing to a new level.

Satellite communication refers to the communication between two or more earth stations by using artificial earth satellites as relay stations to transmit radio waves, and it is a wireless communication technology developed on the basis of microwave communication



and space technology [1]. Satellite network is a special kind of communication network, which has many unique advantages compared with traditional terrestrial networks [2]:

1.  Long distance communication. The cost of communication is independent of distance, so it is particularly suitable for correspondence over the long haul and in areas with few human activities;
2.  Wide communication coverage. Each satellite can cover 42.4% of the global surface, and three GEO satellites can cover the global surface;
3.  Wide communication band and large capacity;
4.  Flexible. Satellite communications can be set up anywhere, regardless of geographical conditions, whether in large cities or remote mountainous areas;
5.  Reliable communication link and strong resistance to natural disasters.

All of these unique merits make satellites play an irreplaceable role when terrestrial communication networks are unavailable or seriously damaged. In recent years, satellite networks have developed rapidly and have become an important pillar in the construction of global information infrastructure. In some countries, satellite Internet has been included in the category of "New Infrastructure" [3].

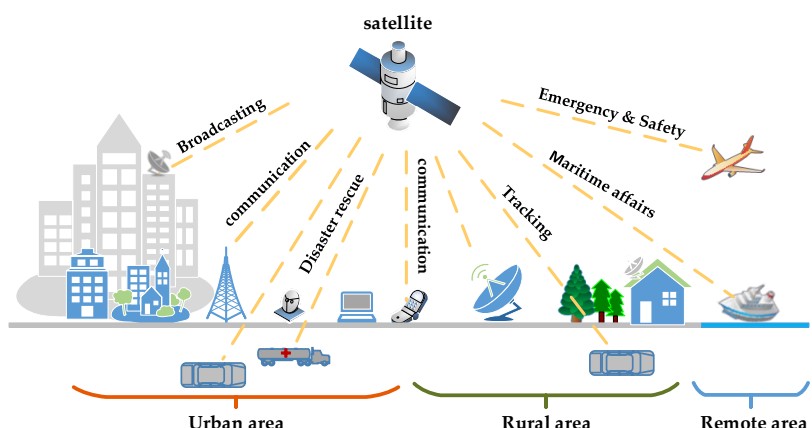

**Figure 1.** Wide application of satellite networks.

In today's life, the increasing level of information technology has brought great convenience to society, but the threat of cyber attacks is also growing. Satellite networks are more vulnerable to the threat of break-ins and attacks due to the openness of its channels. Compared to traditional terrestrial network equipment, satellite equipment is expensive and extremely difficult to repair and recover. Once attacked, it will cause incalculable losses to the country and society [4,5]. More and more scholars have started to pay attention to the security of satellite networks, and build satellite network security protection systems by adopting effective security mechanisms to avoid as much as possible the degradation of network performance or even complete paralysis caused by various attacks. Traditional cyber security techniques such as information encryption, identity authentication and access control can enhance the communication security of satellite networks to some extent, while they can no longer meet the increasingly diversified needs of space tasks. As a result, researchers have begun to explore the application of active defense techniques such as End Hopping (EH) [6,7], Moving Target Defense (MTD) [8], and Mimic Security Defense (MSD) [9] in satellite networks. These strategies create a new way to achieve a shift from a threat-based reactive defense system to a risk-based proactive defense system, and also provide a new idea for space network security protection.

However, network security and performance are often in conflict with each other. Increasing security will lead to degradation of performance metrics (e.g., latency and throughput). Moreover, this situation is even worse in satellite networks, where on-board

resources are severely limited and communication links can be easily blocked. As we all know, "there is no such thing as a free lunch". Active network defense can achieve a high level of security in satellite networks through randomization, dynamization, and diversification, but it is also predictable that the dynamicization mechanism will bring a non-negligible additional load, thus reducing availability of the network. It is necessary for us to have a clear understanding of the cost it brings. Currently, researchers mostly focus on how to further improve the security of satellite networks, and rarely analyze the impact of these defense measures on network performance from a theoretical perspective. This motivates us to evaluate the performance of the satellite networks in a quantitative way.

Aiming to provide theoretical support for rational configuration of active defense strategies and optimization of satellite networks, in this paper, we propose to use SPN to model and analyze the performance of satellite network moving target defense system from a theoretical perspective. We develop SPN models for the communication process of satellite network in the traditional and active protection states, respectively. Then, we evaluate the performance of the satellite network moving target defense system based on the two corresponding SPN models. Through the analysis of models, performance evaluation indexes such as average time delay, throughput, and bandwidth utilization are obtained, and the impact of relevant parameters of active defense technology on network performance is further discussed.

The main contributions of this paper are summarized as follows:

- We propose two scenarios of traditional satellite network and satellite network based on Moving Target Defense technology, respectively;
- We propose a performance evaluation scheme of satellite network moving target defense system based on Stochastic Petri Nets (SPN). We establish SPN models for these two scenarios and evaluate their performance separately;
- We conduct extensive simulations to validate the correctness of theoretical reasoning results and analyze the influence of various factors on the performance indexes of satellite networks. Finally, the challenges and recommendations for deploying active protection technique are given.

The remainder of this paper is organized as follows: in Section 2, we give a brief introduction to Petri Nets and active defense techniques. Then, a literature review of related work is given in Section 3. In Section 4, we establish the SPN models in two scenarios, and then conduct their performance evaluation, respectively. In Section 5, our models are simulated by the PIPE platform, based on which the results of key performance metrics are compared and analyzed, and some specific recommendations are made. In Section 6, we describe the shortcomings of our experiments and the future work. Concluding remarks are given in Section 7.

## 2. Background

In this section, we give some preliminaries. First, a summary of Petri Nets is given, and then a description of Stochastic Petri Nets is provided. Finally, several active defense technologies are presented.

### 2.1. Petri Nets

Petri Nets is a graphical and mathematical modeling tool that can be applied to many systems, and it is also a promising tool to describe and study information processing systems with concurrent, asynchronous, distributed, parallel, uncertain, or random characteristics [10]. Petri Nets has been widely focused on by researchers as soon as it is proposed, and has been widely used in a variety of fields, especially in the analysis and processing of large-scale complex systems. More and more experts and scholars use Petri Net for research. In this paper, we will use this technology to evaluate the performance. In order to make it easier to understand our follow-up work, we present here the relevant knowledge of Petri Nets.

**Definition 1.** *A Petri Net is represented by a five-tuple,* $PN = (P, T, F, W, M_0)$, *where:*

$$P = (P_1, P_2, ..., P_m)$$
$$T = (t_1, t_2, ..., t_n)$$
$$F \subseteq (P \times T) \cup (T \times P)$$
$$W : F \rightarrow \{1, 2, 3, ...\}$$
$$M_0 : P \rightarrow \{0, 1, 2, 3, ...\}$$
$$P \cup T = \oslash \; and \; P \cap T = \oslash$$

$P$ represents the finite set of places, which means the possible local state of the system; $T$ represents the finite set of transitions, which describes the event that can trigger a change in the state of the system; $F$ represents the finite set of directed arcs, which denotes the connection between the state of the system and the event, with the direction either pointing from place to transition or from transition to place. If there exists an arc that goes from a place $P$ to a transition $t$, then $P$ is termed as an input place of $t$. Conversely, if there exists an arc that goes from $t$ to $P$, then $P$ is termed as an output place of $t$. A transition may have one or more input and output places; $W$ is called weight, which is the weight (positive integer) carried on the arc, and is 1 by default; $M_0$ represents the initial marking, understood as the initial state of the whole system, is an m-dimensional vector, m denotes the total number of place $P$, and the p th content of $M$, denoted $M(P)$, represents the number of tokens in the p th place; Tokens are usually contained in places, and can be transferred over different places as the transition occurs. In order to simulate the dynamic behavior of the system, the state or tokens in the Petri Net change according to the following transition (trigger) rules:

1.  If the input position $P$ of each transition $t$ contains at least $W(p, t)$ tokens, where $W(p, t)$ is the weight of the arc from $P$ to $t$, then the transition $t$ is said to be enforceable;
2.  The trigger of an enforceable transition $t$ will result in the removal of $W(p, t)$ tokens from each input place of $t$ and the addition of $W(t, p)$ tokens to each output place of $t$, where $W(t, p)$ is the weight of the arc from $t$ to $P$ .

In the Petri Nets model, places are drawn in the form of circles, transitions are drawn as bars or boxes, and arcs are represented by arcs with arrows; a token is represented by a solid black dot. They are shown in Figure 2.

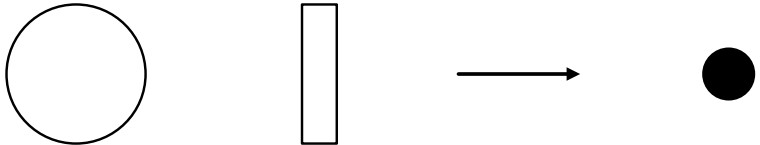

**Figure 2.** Model elements of Petri Nets.

With the continuous study of Petri Nets, researchers have found that the classical PN has many defects. For example, without considering the time factor, the transition will be triggered as soon as the trigger condition is met, i.e., there is no delay, so the time-related performance metrics cannot be obtained. In addition, the scale of the model is easy to become very large, so it is difficult to reason and analyze the model. Therefore, to overcome these shortcomings and describe complex systems more conveniently, many high-level Petri Nets have emerged, including Colored Petri Nets (CPN), Time Petri Nets (TPN), Stochastic Petri Nets (SPN), and so on. Among them, SPN introduces the concept of time into traditional Petri Nets, which is well known for its capability and flexibility in modeling complex systems. Although the dynamic behavior of the model will be affected by the time parameter, the introduction of time reduces the state space and enhances the description ability of Petri Nets. Therefore, in this paper, SPN is used to analyze the performance of satellite networks.

### 2.2. Stochastic Petri Nets

Stochastic Petri Nets (SPN) [11] is extended by Molly on the basis of traditional Petri Nets. By introducing the concept of time, its description ability and solving efficiency are improved, and time-dependent performance metric measures can be obtained as well. Any real-time system can be modeled as a SPN process and can be analyzed by deriving the underlying Markov Chain [12].

**Definition 2.** *A Stochastic Petri Net is represented by a six-tuple, $SPN = (P, T, F, W, M_0, \lambda)$; among them, the meaning of P, T, F, W, $M_0$ is the same as that of traditional Petri Net, $\lambda$ represents the set of average transition firing time rate, indicating the average number of implementations of the transition per unit time under the enforceable case, the average firing time rate corresponding to a transition $t_i$ , i.e., $\lambda_i$. Here,*

$$\lambda = \{\lambda_1, \lambda_2, ... \lambda_n\}$$

SPN introduces a time delay ($\tau_i$) between the non-implementable and implementable of each transition, and the average time delay corresponding to a transition $t_i$ , i.e., $\tau_i$ . Typically, the time delay is assumed to be a continuous random variable with exponential distribution, so it can be isomorphic to a Markov Chain (MC). Furthermore, using the theory of Markov Smooth Distribution, the performance analysis of the system modeled by SPN can be executed by solving the steady-state probability of each marking. This is also the principle of SPN for performance analysis.

The performance evaluation of a system using the SPN model is carried out in the following three steps:

- Step1: Modeling the target system with SPN. This is the first step in conducting a performance evaluation, and the model depends on the concrete system you want to analyze.
- Step2: Constructing the Markov Chain (MC) that is isomorphic to the SPN model.
- Step3: Working on the system performance evaluation with the steady-state probability based on the MC. Specifically, we can use Markov's theory to obtain the steady-state probabilities of each marking and then obtain the relevant parameters.

Here are some formulas to further get those parameters. They are as follows:

1.　Token density function:

$$P[M(P) = i] = \sum_j P(M_j)$$

$$\text{There, } M_j \in [M(P) = i], M_j(P) = i$$

(1)

2.　Average number of tokens on a place:

$$\bar{u}_i = \sum j \times P[M(P_i) = j]$$

(2)

3.　Utilization rate of the transition:

$$U(t_i) = \sum_{M \in E} P(M)$$

(3)

There, $E$ represents the set of all reachable markings that make $t_i$ enforceable.

4.　Token velocity of the transition:

$$R(t_i, P_j) = W(t_i, P_j) \times U(t_i) \times \lambda_i$$

(4)

On the basis of all the performance parameters mentioned above, we can do further research on the average time delay, throughput, and so on.

*2.3. Active Defense Techniques*

As a new technology against cyber attacks, active defense adopts a completely different defense idea from traditional defense techniques, and overcomes the shortcomings of traditional passive defense. Typical active defense technologies include End Hopping (EH), Moving Target Defense (MTD), Mimic Security Defense (MSD), and so on.

EH [6] technology was proposed by Shi in 2008. It refers to military frequency hopping communication countermeasure technology. In the end-to-end data transmission, both sides or one party of the communication pseudorandom change the port, IP address, time slot, protocol, and other End Information according to the agreement to realize the active network defense.

MTD [8] technology is a revolutionary "game-changing" technology in cyberspace proposed by Federal Networking and Information Technology Research and Development (NITRD) in 2011. Unlike prior efforts in cybersecurity research, MTD does not rely on increasing the complexity of the security system to achieve protection of the target. The core idea of MTD is to make the system dynamic, seeking to convert the fixed network into a flexible one, so as to raise the difficulty and cost for attackers and effectively restrict the vulnerabilities exposure and the opportunities for attack.

MSD [9] technology was proposed by Academician Wu in 2014 with reference to the way that mimicry octopus protects itself through morphological changes. The main idea is that, in addition to the service function and performance of the target object not being able to be hidden, the hardware and software of the system can be camouflaged by dynamic changes, so as to achieve the state that the system is controllable to the defender but unknown to the attacker, so as to achieve the purpose of active network defense to protect the system from attack.

According to the above statement, we can clearly know that unlike traditional passive defense methods, active defense techniques are dynamic, versatile, and unpredictable, and are therefore effective in countering direct attacks and interfering with enemy information interception. Among them, MTD is the most representative technology in the active defense system and is a key development direction in the field of future network security. Through the implementation of multi-level, dynamic, and continuous transfer of the attack surface of the protected system, the attacker will have to face as much uncertainty as the defender today, thus reducing the success rate of the intrusion into the system until the attacker is forced to give up the attack. Here, the attack surface can be understood as the set of system resources that can be exploited and attacked in the system. MTD is precisely through the defender to continuously change the resources on the attack surface to achieve changes in the attack surface, so as to confuse or mislead the attacker, prompting the attacker to lose the attack target. At present, the dynamic change technology of the attack surface mainly includes four categories: (1) dynamic change technology based on data attack surface [13]; (2) dynamic change technology based on software attack surface, mainly including instruction set randomization, code randomization [14], etc.; (3) dynamic change technology, based on the platform attack surface, mainly includes platform dynamic migration, virtualization techniques [15], etc.; (4) dynamic change technology based on network attack surface. The main idea is to introduce a dynamic change update mechanism. By collaboratively changing network IP addresses or ports, attackers are always unable to determine the real addresses of the communicating parties, thus undermining the sniffing attacks of attackers and achieving privacy protection for hosts. Among the above four types of attack surface dynamic change technology, the research on MTD based on the change of network attack surface is the most common and mature, and has been widely used.

## 3. Related Work

In recent years, the security of satellite networks has been one of the hot topics. With the popularity of new security technologies of active defense, people's attention to satellite network security has started to shift from traditional protection technologies to active defense techniques.

The first is the network security situation awareness technology proposed by the academic circle, which aims to actively defend against network intrusion behavior and realize network security protection in advance. In [16], the authors introduced situational awareness technology into safety protection of the satellite, and put forward a situational awareness technology system for broadband satellite networks. This research provided support for satellite network security services and also improved the active defense capability of broadband satellite network infrastructure. In [17], the vulnerabilities of space network and the functions of active defense were analyzed, then a simulation implementation method of active defense modeling based on DTN (delay/disruption tolerant network) was proposed. The proposed method can analyze the performance of space network defense system in real time, effectively avoiding "zero-day attacks" and improving the active defense ability of the system. In [18], the authors presented an improved malicious code intrusion detection method for space information network, and the satellite system can achieve fast determination of malicious code attacks. The proposed method has the advantages of high detection rate, low satellite resource consumption, and low latency. In [19], the application design scheme of the endogenous security mechanism of the space-ground integrated information network based on MSD was proposed. This work provided a reference for the construction of a space network active defense security protection system.

Although scholars are gradually exploring the application of active defense technology in satellite communication networks, the related theoretical analysis has not been well studied, so the performance evaluation of satellite network based on active defense has become a necessary and urgent problem. Performance analysis is not only an important theoretical basis and supporting technology for system research, but also an important research direction in various fields. As a powerful analytical tool, PN has been used by scholars to conduct a lot of studies on performance analysis. In the rest of this section, we conduct a literature review on research work based on PNs.

Research on traditional computer network systems and security skills using PNs are very extensive. In [20], the authors presented an efficient Petri-net-based modeling technique for performance evaluation of application mapping. It could precisely represent the exclusion and pipelining of the communication path. The main advantage of this model is the consideration of parallelism of concurrent tasks and communication, as well as the exclusion of computation and communication with public resources. In [21], the authors developed a configurable CPN model for evaluating the performance and the effectiveness of the IEEE 802.11e protocol. Then, they used a CPN model to provide a comprehensive study of the effectiveness of this protocol. Their CPN model sets the basis for further exploring the performance of the various mechanisms defined by the IEEE 802.11 standard. The paper [22] established a performance analysis model based on SPN to evaluate the influence of honeypot on the performance of system. However, this study only stops at concluding whether it is worth deploying honeypots. The impact of honeypots on network performance deserves further study. In [23], a single server retrial queueing system with preemptive priority for modeling and analyzing spectrum occupancy in CR networks was proposed. They analyzed some performance metrics such as delays, throughput, queue length, number of customers in system, etc., via simulation with the help of STCPN (Stochastic Timed Colored Petri Nets). In [24], the authors applied P-Timed Petri Nets to conduct modeling and robustness research on the railway transportation system to evaluate the stability and efficiency of the railway transportation network. Ref. [25] used CPN to model and validate the secure interconnection between industrial control systems (ICS) and enterprise networks. A secure and effective interconnection model between ICS and enterprise networks is proposed, which can be applied to any interconnection environment. There are many other related studies, which will not be listed one by one here.

As for in the area of satellite networks, Petri Nets has been widely used as well, and have accumulated certain research results. Research on the field of satellite network based on PNs mainly include:

In [26], the authors applied Generalized Stochastic Petri Nets (GSPN) to the network control protocols of satellite communication system, and evaluated the reliability and performance of the protocols by verifying some key characteristics of the protocols. Ref. [27] proposed a consistency checking method based on Colored Petri Nets (CPN) to address the possible inconsistency between the protocol specification and the actual protocol execution status in satellite networks. In [28], authors established two Petri Net models to simulate the estimation of space debris flux of different sizes in satellite orbit and to study the impact of debris flux on satellite collision probability prediction, respectively. This is the first work, in our knowledge, to provide a model for a comprehensive evaluation of space debris flux and collision prediction of LEO satellites.

In [29], a navigation satellite availability analysis method based on CPN was proposed. Compared with the traditional availability analysis, this method comprehensively considers the failure factors and performance of the satellite, and is more in line with the actual situation. Ref. [30] proposed an effective reliability assessment algorithm for space information networks based on hopping surface nodes and Petri Nets. This work can guarantee the reliable transmission of data and improve the invulnerability of the network. In [31], the authors proposed a PNs-based method to evaluate the availability of a satellite constellation system. This study can provide guidance for the selection of optimal deployment and maintenance strategies. In [32], the authors simulated a satellite communication network control system based on CPN. When the satellite network communication failure occurs, the system can appropriately reduce the network performance and prevent data loss while maintaining the availability. In [33], the authors proposed a fault diagnosis prototype system of satellite remote control subsystem based on Petri Nets. Compared with the rule-based expert system method, this one can store knowledge in the mathematical matrix and reason more quickly and effectively.

In the field of performance analysis, Ref. [34] studied the data processing effectiveness evaluation of the satellite information application chain. By constructing a Petri Net model, core indicators such as average queue length and average waiting time were analyzed. This research provided support for the optimal allocation of resources in the satellite information application chain. In [35,36], the authors used SPN to construct performance analysis models for the message transmission process of two-layer and three-layer satellite network, respectively. However, the results were not very credible as only the average delay was selected as the network performance evaluation index. In [37], the authors proposed a SPN-based quantitative model for vulnerability, uncertainty, and probability (VUP) of satellite interactive networks. Then, the probability of the network at a given time and the vulnerability and uncertainty of the system under given conditions were calculated and analyzed. In [38], SPN performance evaluation models of the LEO satellite network in the case of single-user and dual-user were established separately. The authors concluded from the theoretical analysis that the satellite network under dual-users makes the average delay greater due to the presence of resource competition. Ref. [39] modeled each operating phase of the microsatellite system separately based on Time Petri Nets (TPN). This research work was very interesting, but unfortunately the results were not very exploitable.

To facilitate a quick overview of these research works, Table 1 summarizes the literature presented in this section.

**Table 1.** Summary of research literature based on Petri Nets.

| Research Field | Reference | Method | Content |
|---|---|---|---|
| Traditional Network | Ref. [20] | PN | Application mapping assessment |
| | Ref. [21] | CPN | Evaluation of IEEE802.11e protocol |
| | Ref. [22] | SPN | Performance evaluation of Honeypot technology |
| | Ref. [23] | STCPN | Modeling of CR networks |
| | Ref. [24] | PTPN | Robustness research on railway transportation |
| | Ref. [25] | CPN | Network security interconnection verification |
| Protocol Correctness Verification | Ref. [26] | GSPN | Evaluate the reliability and performance of the protocols |
| | Ref. [27] | CPN | Protocol consistency check |
| | Ref. [28] | PN | LEO satellite collision prediction |
| Availability Analysis | Ref. [29] | CPN | Navigation satellite availability analysis |
| | Ref. [30] | PN | Reliability assessment |
| | Ref. [31] | PN | Satellite constellation system availability analysis |
| Fault Detection | Ref. [32] | CPN | Simulation of satellite communication network control system |
| | Ref. [33] | PN | Fault diagnosis of satellite remote control subsystem |
| Performance Evaluation | Ref. [34] | PN | Data processing effectiveness evaluation of satellite information application chain |
| | Ref. [35] | SPN | Two-layer satellite network performance analysis |
| | Ref. [36] | SPN | Three-layer satellite network performance analysis |
| | Ref. [37] | SPN | Calculation of vulnerability and uncertainty |
| | Ref. [38] | SPN | Performance comparison between single user and dual user |
| | Ref. [39] | TPN | Modeling of microsatellite system |

In summary, researchers have performed a wide range of applications in protocol correctness verification, availability analysis, fault detection, and performance evaluation of satellite networks based on Petri Net. However, in terms of performance evaluation, the aforementioned works are almost focused on the satellite network itself, application protocols or its networking mode. As far as we know, there is no research on performance evaluation of satellite networks based on active defense, and there is a lack of theoretical guidance for the deployment of proactive security mechanisms. Consequently, in this paper, we propose a performance evaluation scheme of satellite network moving target defense system based on SPN.

## 4. SPN-Based Modeling and Performance Analysis

In this section, to investigate the impact of MTD on the performance of satellite networks, we give two satellite network communication scenarios based on conventional state and moving target defense state separately. Furthermore, we describe our proposed SPN models and demonstrate theoretical reasoning in detail.

### 4.1. Modeling and Analysis of Traditional Satellite Network with SPN

As shown in Figure 3, a typical satellite network communication system usually consists of two parts: the space segment and the ground segment.

The space segment consists of all satellites in the outer space of the Earth. The ground segment can be divided into two parts: the ground base station (including ground uplink station, ground receiving station, measurement and control station, etc.) and the user segment. The former is responsible for the communication access of user terminals in the communication service area, and undertakes the interface function between the satellite communication system and the terrestrial communication network (such as public telephone exchange network, public exchange data network, Internet). The user segment includes all kinds of user terminals (vehicle terminals, ship terminals, etc.) with communication needs. The communication satellite is the core of the entire system. Its main function is to act as a relay station connecting two or more terrestrial base stations, amplifying and relaying signals from terrestrial or other satellites, thus providing rich and colorful communication services with worldwide coverage and meeting the communication needs of users. The links connecting these devices are called communication links, including

uplink, downlink, and inter-satellite link. In satellite communication, the link from the earth station to the satellite is an uplink, and, conversely, the link from the satellite to the earth station is a downlink. The inter-satellite link refers to the radio link connecting satellites. Its function is to connect individual satellites into a space-based network to realize the intercommunication between them, and thus enable the collection, processing, transmission, and distribution of information.

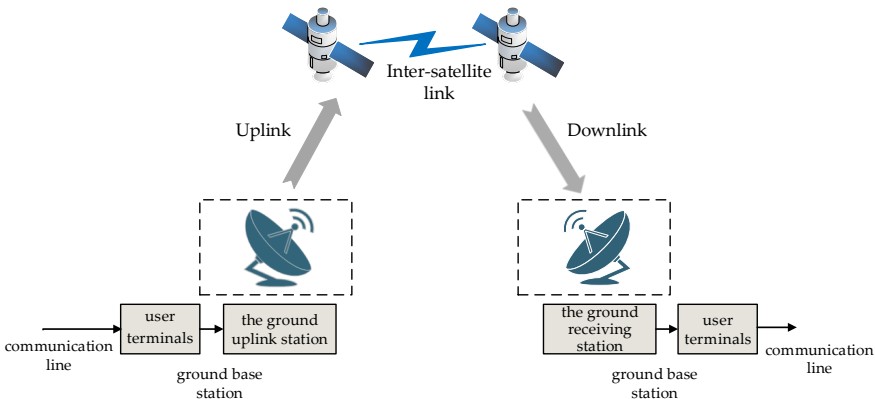

**Figure 3.** The composition of the satellite communication system.

The typical communication process for satellite network is shown in Figure 4. More specifically, it can be described as follows:

1. The user terminal sends a service request to the satellite L1 through base station and waits for the service response;
2. Obtaining link bandwidth resources and L1 responds to the service request;
3. Sending data to L1 through the uplink;
4. L1 forwards the received data via the inter-satellite link to L2, which is responsible for the communication of user segment B;
5. L2 transmits data to B via the downlink, and finally completes the communication between A and B.

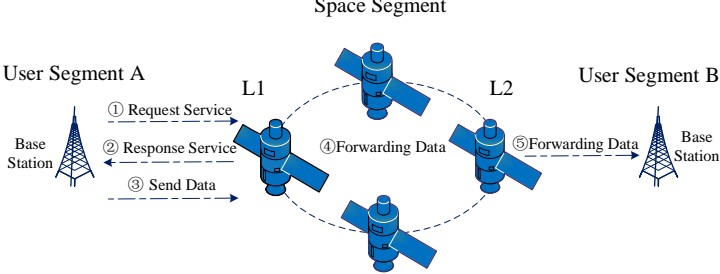

**Figure 4.** Communication process for the satellite network.

We use SPN to study the process by which messages are generated from users on the ground, then transmitted through the satellite network and finally returned to the ground. According to the above communication process and referring to the model in [38], we construct an SPN model of the traditional satellite network communication process, as shown in Figure 5.

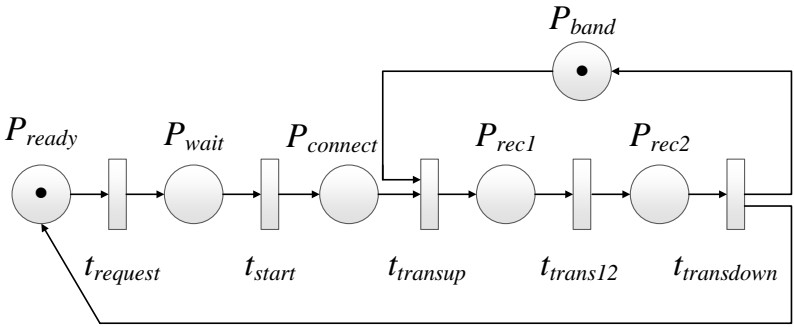

**Figure 5.** SPN model for the traditional satellite network.

The meaning of each place and transition in the SPN model above is listed in Table 2. We denote $\lambda = \{\lambda_2, \lambda_3, \lambda_6, \lambda_7, \lambda_8\}$ as the average transition firing time rate and $\tau = \{\tau_2, \tau_3, \tau_6, \tau_7, \tau_8\}$ as the average implementation delay. The average transition firing time rate and average implementation delay corresponding to each transition can be clearly found from Table 2.

**Table 2.** List of SPN objects in Figure 5.

| Place/Transition | Marking/Rate | Meaning |
|---|---|---|
| $P_{ready}$ | 1 | Ready |
| $P_{wait}$ | 0 | Waiting for transmission link |
| $P_{connect}$ | 0 | Communication/Connection |
| $P_{rec1}$ | 0 | Message arriving to satellite L1 |
| $P_{rec2}$ | 0 | Message arriving to satellite L2 |
| $P_{band}$ | 1 | On-Star Bandwidth Resources |
| $t_{request}$ | $\lambda_2$ | Requesting service |
| $t_{start}$ | $\lambda_3$ | Starting service |
| $t_{transup}$ | $\lambda_6$ | Transmitting message from ground to satellite L1 via uplink |
| $t_{trans12}$ | $\lambda_7$ | Transmitting message from L1 to L2 via inter-satellite link |
| $t_{transdown}$ | $\lambda_8$ | Transmitting message from L2 to ground via downlink |

According to the performance evaluation process in [40], the performance of the established SPN model can be analyzed by using Petri Nets theory and Markov theory. First, we can get the reachable marking set as $M = \{ M_0, M_1, M_2, M_3, M_4 \}$ of the traditional satellite network SPN model, as shown in Table 3.

**Table 3.** Reachable marking set of the SPN model.

| | $P_{ready}$ | $P_{wait}$ | $P_{connect}$ | $P_{rec1}$ | $P_{rec2}$ | $P_{band}$ |
|---|---|---|---|---|---|---|
| $M_0$ | 1 | 0 | 0 | 0 | 0 | 1 |
| $M_1$ | 0 | 1 | 0 | 0 | 0 | 1 |
| $M_2$ | 0 | 0 | 1 | 0 | 0 | 1 |
| $M_3$ | 0 | 0 | 0 | 1 | 0 | 0 |
| $M_4$ | 0 | 0 | 0 | 0 | 1 | 0 |

Since the SPN reachable graph is isomorphic to a Continuous Time Markov Chain (CTMC), the isomorphic MC can be obtained by replacing each transition in the reachable graph with its corresponding average firing time rate as shown in Figure 6 [40]. The isomorphic MC contains five markings: $M_0(100001)$, $M_1(010001)$, $M_2(001001)$, $M_3(000100)$, $M_4(000010)$.

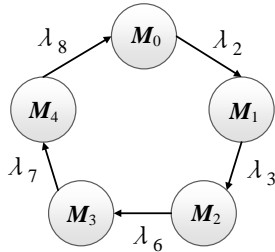

**Figure 6.** Isomorphic Markov chain.

According to the definition of the transition matrix: $Q = [q_{i,j}]$, $i \leq n\ j \leq n$, there:

$$q_{i,j} = \begin{cases} \lambda_k, \text{the rate on the arc from } M_i \text{ to} M_j \text{ when } i \neq j \\ 0, \text{no arc from } M_i \text{ to } M_j \text{ when } i \neq j \\ -\sum_k \lambda_k, \ i = j \end{cases} \tag{5}$$

We can estimate the SPN model as follows, the transition matrix $Q$ of the SPN model is:

$$Q = \begin{bmatrix} -\lambda_2 & \lambda_2 & 0 & 0 & 0 \\ 0 & -\lambda_3 & \lambda_3 & 0 & 0 \\ 0 & 0 & -\lambda_6 & \lambda_6 & 0 \\ \lambda_2 & 0 & 0 & -\lambda_7 & \lambda_7 \\ \lambda_8 & 0 & 0 & 0 & -\lambda_8 \end{bmatrix}$$

We assume that $P = (p_0, p_1, p_2, p_3, p_4)$, $p_i$ represents the steady-state probability of the Marking $M_i$. According to Markovian Smooth Distribution theory, we have:

$$\begin{cases} P \times Q = 0 \\ \sum_0^4 p_i = 1 \end{cases} \tag{6}$$

By solving the linear equation system from (6), the steady-state probability of each marking can be obtained as follows:

$$P(M_0) = p_0 = \lambda_2^{-1}/(\lambda_2^{-1} + \lambda_3^{-1} + \lambda_6^{-1} + \lambda_7^{-1} + \lambda_8^{-1})$$
$$P(M_1) = p_1 = \lambda_3^{-1}/(\lambda_2^{-1} + \lambda_3^{-1} + \lambda_6^{-1} + \lambda_7^{-1} + \lambda_8^{-1})$$
$$P(M_2) = p_2 = \lambda_6^{-1}/(\lambda_2^{-1} + \lambda_3^{-1} + \lambda_6^{-1} + \lambda_7^{-1} + \lambda_8^{-1})$$
$$P(M_3) = p_3 = \lambda_7^{-1}/(\lambda_2^{-1} + \lambda_3^{-1} + \lambda_6^{-1} + \lambda_7^{-1} + \lambda_8^{-1})$$
$$P(M_4) = p_4 = \lambda_8^{-1}/(\lambda_2^{-1} + \lambda_3^{-1} + \lambda_6^{-1} + \lambda_7^{-1} + \lambda_8^{-1})$$

Furthermore, by applying the steady-state probabilities of markings and Formulas (1)–(4) in Section 2.2, the basic performance metrics such as token density function in each place, average number of tokens on a place, utilization rate of the transition, token velocity of the transition, etc. can be derived easily.

1.  Token density function in each place is as follows:

$$P(M(P_{ready}) = 1) = P(M_0) = p_0$$
$$P(M(P_{wait}) = 1) = P(M_1) = p_1$$
$$P(M(P_{connect}) = 1) = P(M_2) = p_2$$
$$P(M(P_{rec1}) = 1) = P(M_3) = p_3$$
$$P(M(P_{rec2}) = 1) = P(M_4) = p_4$$
$$P(M(P_{band}) = 1) = P(M_0) + P(M_1) + p(M_2) = p_0 + p_1 + p_2$$

2.  The average number of tokens on a place in the steady-state can be calculated as:

$$\bar{u}_{ready} = 1 \times P(M(P_{ready}) = 1) = p_0$$
$$\bar{u}_{wait} = 1 \times P(M(P_{wait}) = 1) = p_1$$
$$\bar{u}_{connect} = 1 \times P(M(P_{connect}) = 1) = p_2$$
$$\bar{u}_{rec1} = 1 \times P(M(P_{rec1}) = 1) = p_3$$
$$\bar{u}_{rec2} = 1 \times P(M(P_{rec2}) = 1) = p_4$$
$$\bar{u}_{band} = 1 \times P(M(P_{band}) = 1) = p_0 + p_1 + p_2$$

The average number of tokens contained in the set of all places from the time the service is requested by the user to the time it is completed is calculated as:

$$\overline{N} = \bar{u}_{wait} + \bar{u}_{connect} + \bar{u}_{rec1} + \bar{u}_{rec2} + \bar{u}_{band} = 1 + p_1 + p_2$$

3. The utilization rate of $t_{request}$ is:

$$U(t_{requeest}) = P(M_0) = p_0$$

4. The rate from $t_{request}$ to $P_{wait}$ is:

$$R(t_{request}, P_{wait}) = W(t_{request}, P_{wait}) \times U(t_{request}) \times \lambda_2 = \lambda_2 p_0$$

On the basis of all the performance parameters mentioned above, we can further calculate the average time delay by applying Little's theorem and principle of balance [41], Little's theorem is formulated as:

$$\overline{N} = \lambda T \tag{7}$$

$\overline{N}$ represents the average queue length, $\lambda$ denotes the average arrival rate, and $T$ means average time delay of the queue. Consequently, the average time delay of the network is:

$$T = \overline{N}/R(t_{request}, P_{wait}) = (1 + p_1 + p_2)/\lambda_2 p_0 = \lambda_2^{-1} + 2\lambda_3^{-1} + 2\lambda_6^{-1} + \lambda_7^{-1} + \lambda_8^{-1} \tag{8}$$

The average throughput is defined as the average number of tasks completed by the system per unit of time, and it is an important indicator characterizing the performance of the system. The formula for calculating the throughput of each transition $t$ in the steady-state is:

$$O(t) = \sum_{M \in H} P(M) \times \lambda_t \tag{9}$$

$H$ is the set of markings that enable the implementation of transition $t$, and $\lambda_t$ is the average firing time rate of the transition $t$ under marking $M$. The SPN model of satellite network communication completes one data communication service after the implementation of transition $t_{transdown}$. Therefore, the average system throughput is:

$$O = P(M_4) \times \lambda_8 = \lambda_8 p_4 = \frac{1}{\lambda_2^{-1} + \lambda_3^{-1} + \lambda_6^{-1} + \lambda_7^{-1} + \lambda_8^{-1}} \tag{10}$$

Utilization of on-board bandwidth resources, which is used to measure the consumption of bandwidth resources on satellite networks, is calculated as:

$$U = P(M(P_{band}) = 0) = p_3 + p_4 = \frac{\lambda_7^{-1} + \lambda_8^{-1}}{\lambda_2^{-1} + \lambda_3^{-1} + \lambda_6^{-1} + \lambda_7^{-1} + \lambda_8^{-1}} \tag{11}$$

Since the stochastic process used in the SPN model is a Poisson process, the occurrence of the transition satisfies the Poisson distribution, so the reciprocal of the average firing time rate of each transition in the model is its average implementation delay, i.e., $\tau_i = 1/\lambda_i$. To facilitate the analysis of the factors affecting each performance indicators, the following explanation is given: $\lambda_2^{-1}$ denotes request delay $\tau_2$, $\lambda_3^{-1}$ denotes waiting service delay $\tau_3$, $\lambda_6^{-1}$, $\lambda_7^{-1}$, and $\lambda_8^{-1}$ denotes uplink propagation delay $\tau_6$, inter-satellite link propagation delay $\tau_7$, and downlink propagation delay $\tau_8$, respectively.

It can be seen that the average delay and throughput of satellite network communication process are closely related to the request duration, service latency, propagation delay of uplink, downlink, and inter-satellite link. The average network delay is the sum of each process delays, while the throughput rate is inversely proportional to this, which is consistent with the actual situation. As a result, in traditional satellite networks, accelerating service response, processing speed, and improving the transmission efficiency of the links between users and satellites can effectively reduce the average network delay, increase the throughput, and improve the network performance.

### 4.2. Modeling and Analysis of Active Defense-Based Satellite Network with SPN

From Section 2.3, we know that the research on MTD based on the change of network attack surface is the most common and mature. Therefore, this paper focuses on the modeling and analysis of the satellite network moving target defense system based on the change of network attack surface. The communication scenario is shown in Figure 7.

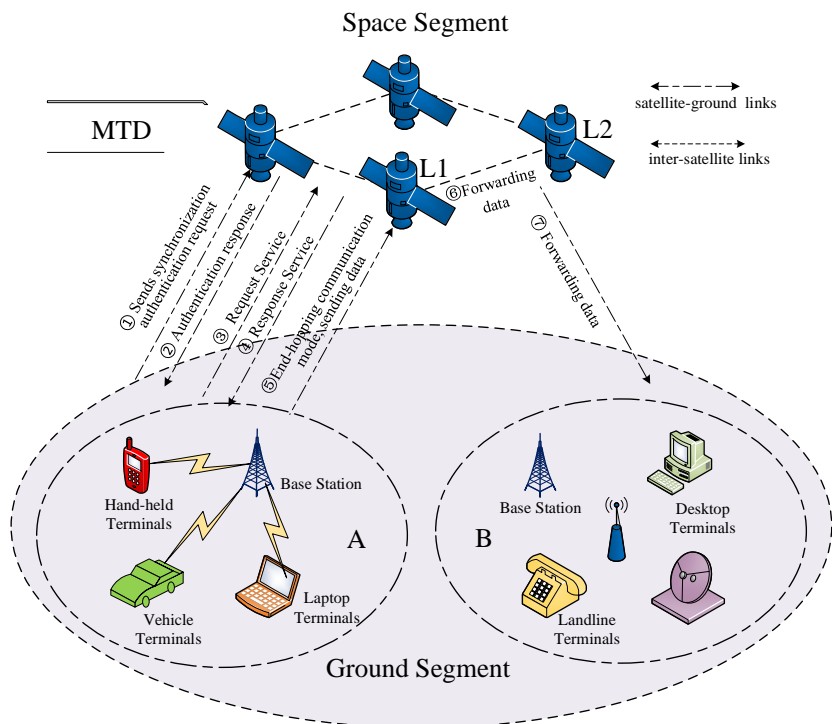

**Figure 7.** Communication scenario for the satellite network security protection system based on Moving Target Defense.

When there is a communication demand between two user segments A and B, the MTD-based satellite network communication process is as follows:

1. The ground user terminal in user segment A sends a synchronization authentication request to satellite L1 according to the established rules;
2. L1 responds to the user with authentication and turns on the synchronization service to ensure the legitimacy of the access entity and to resist spoofing by unauthorized users;
3. The authenticated trusted user terminal sends a service request to L1 and waits for the service response;
4. Both parties obtain link bandwidth resources and L1 responds to the service request;

5. Both parties switch to Moving Target Defense communication mode, and the client transmits data to L1 through the uplink. Whenever the current satellite service time slot ends, the service provider performs hopping (IP Address, Port) and data migration, then both parties continue the unfinished communication until this uplink message transmission is completed;

6. L1 forwards the received data via the inter-satellite link to satellite L2, which is responsible for the communication of user segment B;

7. L2 transmits the data to B via the downlink, thus completing the communication between A and B.

Based on the above analysis and the study in [42], the corresponding SPN model is obtained as shown in Figure 8. The meaning of each place and transition as well as the corresponding number of tokens is shown in Table 4. We denote $\lambda = \{\lambda_1, \lambda_2, \lambda_3, \lambda_4, \lambda_5, \lambda_6, \lambda_7, \lambda_8\}$ as the average transition firing time rate and $\tau = \{\tau_1, \tau_2, \tau_3, \tau_4, \tau_5, \tau_6, \tau_7, \tau_8\}$ as the average implementation delay. From Table 4, we can see that the corresponding relationship between the transition and the above two parameters is consistent with those described in Section 4.1.

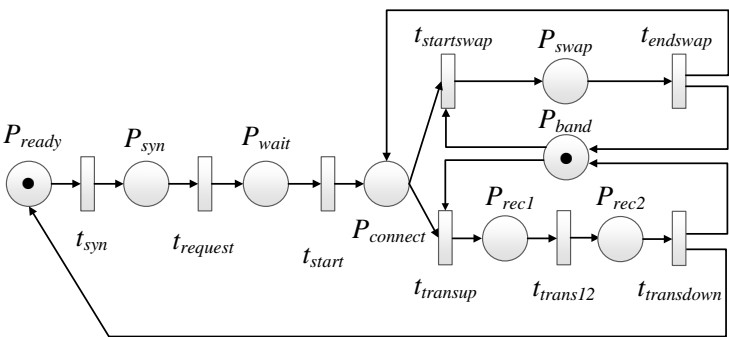

**Figure 8.** SPN model for satellite networks based on Moving Target Defense.

Compared with the model in Figure 5, the SPN model based on Moving Target Defense adds a synchronization place ($P_{syn}$) and a migration place ($P_{swap}$). Different from the traditional satellite network communication process, the user needs to send a synchronous authentication request ($t_{syn}$) before the communication is established, and only after the authentication is successful can the connection, service, and other processes be carried out. During the service process, when the satellite time slot ($\tau_4$) of the current service is exhausted, service hopping, and data migration will be conducted, this process will consume additional bandwidth resources. After all the messages from client have been received for this communication, the uplink transmission ends and the system enters a new state $P_{rec1}$. Through analysis, we can easily get the reachable set of markings as shown in Table 5, and, furthermore, we can construct its isomorphic MC as shown in Figure 9.

**Table 4.** List of SPN objects in Figure 8.

| Place/Transition | Marking/Rate | Meaning |
|---|---|---|
| $P_{ready}$ | 1 | Ready |
| $P_{syn}$ | 0 | Synchronization |
| $P_{wait}$ | 0 | Waiting for transmission link |
| $P_{connect}$ | 0 | Communication/Connection |
| $P_{swap}$ | 0 | Migration / Service switching |
| $P_{rec1}$ | 0 | Message arriving to satellite L1 |
| $P_{rec2}$ | 0 | Message arriving to satellite L2 |
| $P_{band}$ | 1 | On-Star Bandwidth Resources |
| $t_{syn}$ | $\lambda_1$ | Requesting synchronization |
| $t_{request}$ | $\lambda_2$ | Requesting service |
| $t_{start}$ | $\lambda_3$ | Starting service |
| $t_{startswap}$ | $\lambda_4$ | Staring swap |
| $t_{endswap}$ | $\lambda_5$ | Ending swap |
| $t_{transup}$ | $\lambda_6$ | Transmitting message from ground to satellite L1 via uplink |
| $t_{trans12}$ | $\lambda_7$ | Transmitting message from L1 to L2 via inter-satellite link |
| $t_{transdown}$ | $\lambda_8$ | Transmitting message from L2 to ground via downlink |

**Table 5.** Reachable marking set of the SPN model.

| | $P_{ready}$ | $P_{syn}$ | $P_{wait}$ | $P_{connect}$ | $P_{swap}$ | $P_{rec1}$ | $P_{rec2}$ | $P_{band}$ |
|---|---|---|---|---|---|---|---|---|
| $M_0$ | 1 | 0 | 0 | 0 | 0 | 0 | 0 | 1 |
| $M_1$ | 0 | 1 | 0 | 0 | 0 | 0 | 0 | 1 |
| $M_2$ | 0 | 0 | 1 | 0 | 0 | 0 | 0 | 1 |
| $M_3$ | 0 | 0 | 0 | 1 | 0 | 0 | 0 | 1 |
| $M_4$ | 0 | 0 | 0 | 0 | 1 | 0 | 0 | 0 |
| $M_5$ | 0 | 0 | 0 | 0 | 0 | 1 | 0 | 0 |
| $M_6$ | 0 | 0 | 0 | 0 | 0 | 0 | 1 | 0 |

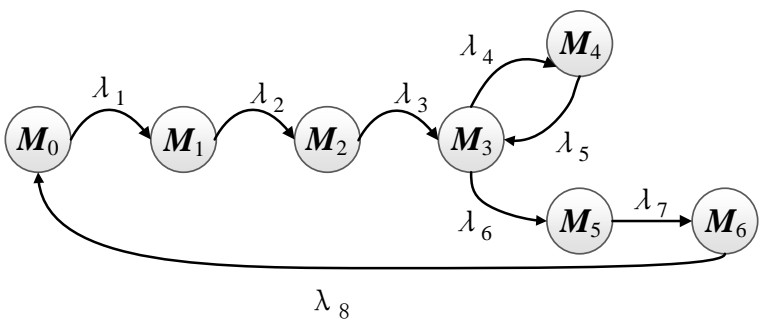

**Figure 9.** Isomorphic Markov chain.

The isomorphic MC contains seven markings: $M_0(10000001)$, $M_1(01000001)$, $M_2(00100001)$, $M_3(00010001)$, $M_4(00001000)$, $M_5(00000100)$, $M_6(00000010)$, the corresponding transition matrix $Q$ of this SPN model is:

$$Q = \begin{bmatrix} -\lambda_1 & \lambda_1 & 0 & 0 & 0 & 0 & 0 \\ 0 & -\lambda_2 & \lambda_2 & 0 & 0 & 0 & 0 \\ 0 & 0 & -\lambda_3 & \lambda_3 & 0 & 0 & 0 \\ 0 & 0 & 0 & -\lambda_4 - \lambda_6 & \lambda_4 & \lambda_6 & 0 \\ 0 & 0 & 0 & -\lambda_5 & \lambda_5 & 0 & 0 \\ 0 & 0 & 0 & 0 & 0 & -\lambda_7 & \lambda_7 \\ \lambda_8 & 0 & 0 & 0 & 0 & 0 & -\lambda_8 \end{bmatrix}$$

Similarly, let $P = (p_0, p_1, p_2, p_3, p_4, p_5, p_6)$ be the row vector corresponding to the steady-state probability of each marking, and according to Equation (6), the steady-state probability of each marking is obtained as:

$$P(M_0) = p_0 = \lambda_1^{-1} / (\lambda_1^{-1} + \lambda_2^{-1} + \lambda_3^{-1} + \lambda_6^{-1} + \lambda_5^{-1}\lambda_4\lambda_6^{-1} + \lambda_7^{-1} + \lambda_8^{-1})$$
$$P(M_1) = p_1 = \lambda_2^{-1} / (\lambda_1^{-1} + \lambda_2^{-1} + \lambda_3^{-1} + \lambda_6^{-1} + \lambda_5^{-1}\lambda_4\lambda_6^{-1} + \lambda_7^{-1} + \lambda_8^{-1})$$
$$P(M_2) = p_2 = \lambda_3^{-1} / (\lambda_1^{-1} + \lambda_2^{-1} + \lambda_3^{-1} + \lambda_6^{-1} + \lambda_5^{-1}\lambda_4\lambda_6^{-1} + \lambda_7^{-1} + \lambda_8^{-1})$$
$$P(M_3) = p_3 = \lambda_6^{-1} / (\lambda_1^{-1} + \lambda_2^{-1} + \lambda_3^{-1} + \lambda_6^{-1} + \lambda_5^{-1}\lambda_4\lambda_6^{-1} + \lambda_7^{-1} + \lambda_8^{-1})$$
$$P(M_4) = \lambda_5^{-1}\lambda_4\lambda_6^{-1} / (\lambda_1^{-1} + \lambda_2^{-1} + \lambda_3^{-1} + \lambda_6^{-1} + \lambda_5^{-1}\lambda_4\lambda_6^{-1} + \lambda_7^{-1} + \lambda_8^{-1})$$
$$P(M_5) = p_5 = \lambda_7^{-1} / (\lambda_1^{-1} + \lambda_2^{-1} + \lambda_3^{-1} + \lambda_6^{-1} + \lambda_5^{-1}\lambda_4\lambda_6^{-1} + \lambda_7^{-1} + \lambda_8^{-1})$$
$$P(M_6) = p_6 = \lambda_8^{-1} / (\lambda_1^{-1} + \lambda_2^{-1} + \lambda_3^{-1} + \lambda_6^{-1} + \lambda_5^{-1}\lambda_4\lambda_6^{-1} + \lambda_7^{-1} + \lambda_8^{-1})$$

1.  Token density function in each place is as follows:

$$P(M(P_{ready}) = 1) = P(M_0) = p_0$$
$$P(M(P_{syn}) = 1) = P(M_1) = p_1$$
$$P(M(P_{wait}) = 1) = P(M_2) = p_2$$
$$P(M(P_{connect}) = 1) = P(M_3) = p_3$$
$$P(M(P_{swap}) = 1) = P(M_4) = p_4$$
$$P(M(P_{rec1}) = 1) = P(M_5) = p_5$$
$$P(M(P_{rec2}) = 1) = P(M_6) = p_6$$
$$P(M(P_{band}) = 1) = P(M_0) + P(M_1) + p(M_2) + p(M_3) = p_0 + p_1 + p_2 + p_3$$

2.  The average number of tokens on a place in the steady-state can be calculated as:

$$\bar{u}_{ready} = p_0, \ \bar{u}_{syn} = p_1$$
$$\bar{u}_{wait} = p_2, \bar{u}_{connect} = p_3$$
$$\bar{u}_{swap} = p_4, \ \bar{u}_{rec1} = p_5$$
$$\bar{u}_{rec2} = p_6, \bar{u}_{band} = p_0 + p_1 + p_2 + p_3$$

The average number of tokens contained in the set of all places from the request made by the client to the completion of the service is calculated as:

$$\overline{N} = \bar{u}_{wait} + \bar{u}_{swap} + \bar{u}_{connect} + \bar{u}_{rec1} + \bar{u}_{rec2} + \bar{u}_{band} = 1 + p_1 + p_2 + p_3$$

3.  The utilization rate of $t_{request}$ is:

$$U(t_{requeest}) = P(M_1) = p_1$$

4.  The rate from $t_{request}$ to $P_{wait}$ is:

$$R(t_{request}, P_{wait}) = W(t_{request}, P_{wait}) \times U(t_{request}) \times \lambda_2 = \lambda_2 p_1$$

Therefore, the average latency of the satellite network security protection system based on the Moving Target Defense technique is:

$$T = \overline{N} / R(t_{request}, P_{wait}) = \lambda_1^{-1} + \lambda_2^{-1} + 2\lambda_3^{-1} + 2\lambda_6^{-1} + \lambda_5^{-1}\lambda_4\lambda_6^{-1} + \lambda_7^{-1} + \lambda_8^{-1} \quad (12)$$

Average throughput is:

$$O = P(M_6) \times \lambda_8 = \lambda_8 p_6 = \frac{1}{\lambda_1^{-1} + \lambda_2^{-1} + \lambda_3^{-1} + \lambda_6^{-1} + \lambda_5^{-1}\lambda_4\lambda_6^{-1} + \lambda_7^{-1} + \lambda_8^{-1}} \quad (13)$$

Utilization of on-board bandwidth resources is:

$$U = P(M(P_{band}) = 0) = \frac{\lambda_5^{-1}\lambda_4\lambda_6^{-1} + \lambda_7^{-1} + \lambda_8^{-1}}{\lambda_1^{-1} + \lambda_2^{-1} + \lambda_3^{-1} + \lambda_6^{-1} + \lambda_5^{-1}\lambda_4\lambda_6^{-1} + \lambda_7^{-1} + \lambda_8^{-1}} \quad (14)$$

$\lambda_2^{-1}, \lambda_3^{-1}, \lambda_6^{-1}, \lambda_7^{-1}, \lambda_8^{-1}$ have the same meaning as described in Section 4.1, $\lambda_1^{-1}$ denotes the synchronous authentication delay i.e., $\tau_1$ , $\lambda_4^{-1}$ denotes the single service hopping time slot i.e., $\tau_4$ , and $\lambda_5^{-1}$ denotes the data migration delay i.e., $\tau_5$ .

From the theoretical reasoning results, we can see that the network time delay, throughput, and bandwidth utilization of the satellite network moving target defense system are

not only related to the link transmission duration, request delay, and service delay, but also affected by synchronization delay, hopping rate, and data migration delay introduced by MTD strategy. Compared with traditional satellite networks, the MTD-based satellite network has the following relationship in terms of average delay, average throughput, and bandwidth utilization:

$$
\lambda_1^{-1} + \lambda_2^{-1} + 2\lambda_3^{-1} + 2\lambda_6^{-1} + \lambda_5^{-1}\lambda_4\lambda_6^{-1} + \lambda_7^{-1} + \lambda_8^{-1} \geq \lambda_2^{-1} + 2\lambda_3^{-1} + 2\lambda_6^{-1} + \lambda_7^{-1} + \lambda_8^{-1}
$$

$$
\frac{1}{\lambda_1^{-1} + \lambda_2^{-1} + 2\lambda_3^{-1} + 2\lambda_6^{-1} + \lambda_5^{-1}\lambda_4\lambda_6^{-1} + \lambda_7^{-1} + \lambda_8^{-1}} \leq \frac{1}{\lambda_2^{-1} + 2\lambda_3^{-1} + 2\lambda_6^{-1} + \lambda_7^{-1} + \lambda_8^{-1}} \tag{15}
$$

$$
\frac{\lambda_5^{-1}\lambda_4\lambda_6^{-1} + \lambda_7^{-1} + \lambda_8^{-1}}{\lambda_1^{-1} + \lambda_2^{-1} + 2\lambda_3^{-1} + 2\lambda_6^{-1} + \lambda_5^{-1}\lambda_4\lambda_6^{-1} + \lambda_7^{-1} + \lambda_8^{-1}} \leq \frac{\lambda_7^{-1} + \lambda_8^{-1}}{\lambda_2^{-1} + 2\lambda_3^{-1} + 2\lambda_6^{-1} + \lambda_7^{-1} + \lambda_8^{-1}}
$$

Consequently, the synchronous authentication technology, service hopping, and data migration introduced by MTD will reduce the overall performance of the satellite network, specifically, increase the average network latency, reduce the average throughput, and increase the consumption of bandwidth resources on the satellite network.

## 5. Model Simulation and Experiments

In order to verify the impact of MTD technology on satellite network performance, and to explore the variation of average delay, average throughput, and on-star bandwidth resource utilization with different parameters, in this section, simulations are performed for proposed models and a large number of experiments are carried out for the performance analysis work.

First, the two proposed SPN models are simulated separately based on PIPE (Platform Independent Petri Net Editor), one of the simulation tools for Petri Nets, which can draw Petri Net models, simulate the dynamic effects of Petri Nets, and can verify the correctness and usability of the models. The initial reference values of each model are set as shown in Table 6.

**Table 6.** Initial parameters' values.

| Parameters | Values | Parameters | Values |
|:---:|:---:|:---:|:---:|
| $\lambda_1$ | 2 | $\lambda_5$ | 2 |
| $\lambda_2$ | 1 | $\lambda_6$ | 0.6 |
| $\lambda_3$ | 0.4 | $\lambda_7$ | 0.8 |
| $\lambda_4$ | 2 | $\lambda_8$ | 0.6 |

Figures 10 and 11 show the reachable graphs obtained from the simulation of the two SPN models, respectively. By replacing each transition with its corresponding average firing time rate, the same isomorphic Markov Chain as in the theoretical analysis can be obtained, as shown in Figures 6 and 9, where $S_i$ corresponds to $M_i$, thus verifying the correctness of the theoretical inference results. Furthermore, based on the initial parameter values in Table 6, the simulation results of the steady-state probabilities of each marking are shown in Table 7.

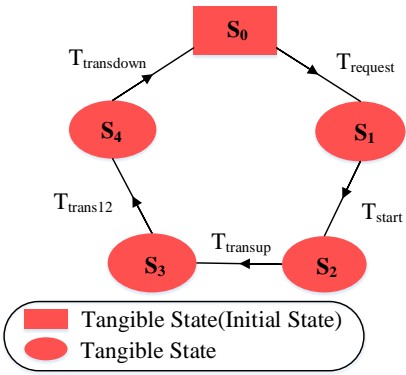

**Figure 10.** Reachability diagram of SPN model 1.

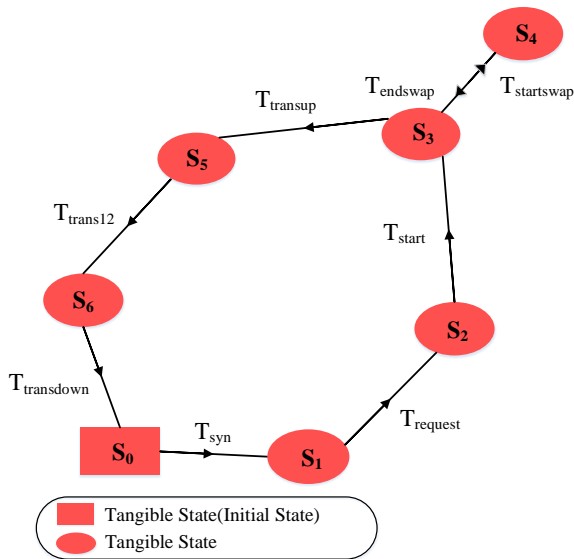

**Figure 11.** Reachability diagram of SPN model 2.

**Table 7.** Steady-state probabilities of each marking.

| Steady-State Probabilities | SPN Model 1 | SPN Model 2 |
|:---:|:---:|:---:|
| $P(M_0)$ | 0.12371 | 0.04878 |
| $P(M_1)$ | 0.30928 | 0.09756 |
| $P(M_2)$ | 0.20619 | 0.2439 |
| $P(M_3)$ | 0.15464 | 0.1626 |
| $P(M_4)$ | 0.20619 | 0.1626 |
| $P(M_5)$ | — | 0.12195 |
| $P(M_6)$ | — | 0.1626 |

We use the data obtained from the above simulations and the formulas in Section 4 to calculate the average delay $T$, average throughput $O$, and bandwidth utilization $U$ of the network under each scenario. The results are shown in Table 8.

**Table 8.** Results of performance indicators.

| Indicators | SPN Model 1 | SPN Model 2 |
|:---:|:---:|:---:|
| $T$ (Average Time Delay) | 12.250 | 14.417 |
| $O$ (Average Throughput) | 0.124 | 0.098 |
| $U$ (Utilization of Bandwidth) | 0.361 | 0.447 |

We can see from Table 8 that the introduction of the MTD mechanism has led to a decline in the performance of the satellite network system, which is consistent with the theoretical reasoning results. More specifically, the average delay and bandwidth resource utilization have increased by 17% and 23%, respectively, while the average throughput of the system has decreased by about 20%. To illustrate the influence relationship of parameters on the selected three performance metrics more clearly and quantitatively, further experiments are conducted by varying values of parameters.

**Experiment 1.** *With the other parameters fixed, the changes of the average delay of the whole network with the increase of synchronization rate, hopping rate, and migration efficiency are investigated, respectively. The results are shown in Figure 12. The horizontal coordinate represents the change rate, and the vertical coordinate represents the average delay.*

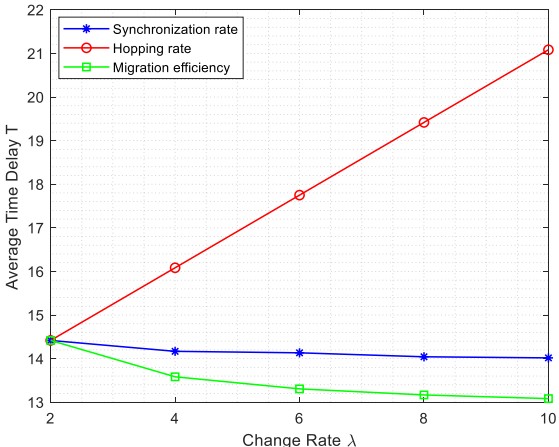

**Figure 12.** Effect of change rate on Average Time Delay.

**Experiment 2.** *Under the condition of Experiment 1, the variation of the average throughput of the whole network is examined. The results are shown in Figure 13, where the horizontal coordinate represents the rate of change and the vertical coordinate stands for the average throughput.*

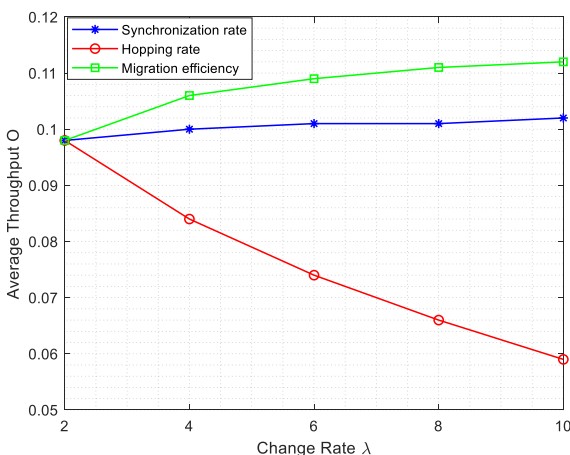

**Figure 13.** Effect of change rate on Average Throughput.

From Figures 12 and 13, we can see that the average delay and average throughput of the network vary drastically with the hopping rate, while the synchronization rate and the data migration efficiency have little effect on them. As the hopping rate increases, the average network throughput decreases significantly, while the latency increases significantly. Since the hopping rate, synchronization rate, and migration efficiency depend on the specific hopping strategy, synchronization technique and migration scheme, respectively,

the hopping strategy plays an important role in delay and throughput. Good synchronization authentication and data migration scheme imply higher synchronization and migration efficiency. Therefore, designing and implementing an efficient synchronization strategy and migration scheme can really improve the latency and average throughput reduction brought by MTD technology to some extent.

**Experiment 3.** *Under the conditions of Experiment 1, the change in bandwidth resource utilization of the whole network is investigated, and the results are shown in Figure 14, where the horizontal coordinate represents the rate of change, and the vertical coordinate represents the bandwidth utilization.*

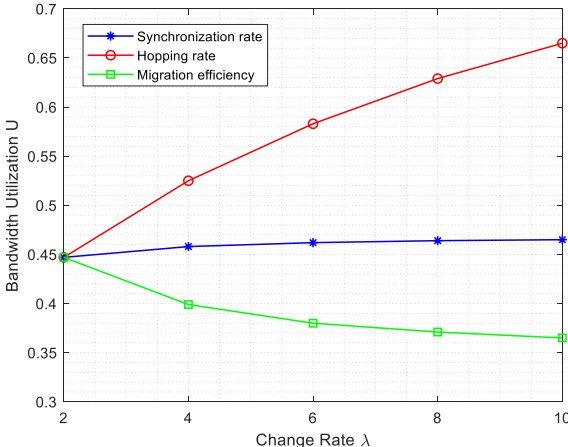

**Figure 14.** Effect of change rate on utilization of bandwidth.

As shown in Figure 14, hopping rate continues to be the main factor affecting the consumption of bandwidth resources. As the hopping rate increases, the bandwidth utilization of the network link also grows. The reasons for this phenomenon can be explained as follows: faster hopping speed, shorter duration of single service, and more frequent data migration, which consumes bandwidth resources additionally. Thus, the competition between normal business and data migration to use network bandwidth makes the consumption of bandwidth resources more apparent. The increase in migration efficiency can reduce the utilization of bandwidth, so an efficient data migration scheme will go some way to alleviating the consumption of bandwidth resources caused by high-speed hopping.

Finally, one more work has been done to illustrate the superiority of our scheme. In the field of performance evaluation, the three most representative mathematical theoretical analysis methods are Queueing Theory, Markov Process, and Petri Nets. We made a horizontal comparison between our method and the work in [43,44]. Results are shown in Table 9.

**Table 9.** Comparison of features of different performance evaluation schemes.

| Feature | Reference [43] | Reference [44] | This Paper |
|---|---|---|---|
| Description ability | Medium | Strong | Strong |
| Modeling ability | Medium | Medium | Strong |
| Portrayal ability | Medium | Strong | Strong |

We can see that the performance analysis method we use has excellent performance in terms of descriptive, modeling, and characterization capabilities. Queueing Theory in [43] has limitations in modeling relatively complex structures and cannot portray the parallel, asynchronous, and distributed characteristics of information operations. The Markov Process of [44] performs slightly worse in modeling and has difficulty in modeling the corresponding stochastic process level.

Summary: the deployment and implementation of MTD mechanism in satellite networks can effectively improve the security of the network on the one hand. On the other hand, it also brings performance loss. Hopping rate is the key factor influencing network performance. Extending to the category of active defense technology, which is typically characterized by "proactive change," a high frequency of change is required to keep the system dynamic and defensive. Therefore, to deploy active defense technology in the special environment of satellite networks, the setting of change frequency is the key bottleneck, and it is necessary to comprehensively consider the network environment and security requirements in practical applications, so as to obtain availability–security–overhead balance. In addition, it is crucial to study efficient satellite-ground synchronization authentication technology and data migration scheme, and to improve the transmission efficiency of satellite-ground and inter-satellite links, and to improve the service response as well as satellite processing business capacity, in order to reduce the impact of active protection technology on network performance and to achieve low overhead processing capability of satellite networks.

## 6. Discussion

With the gradual application of active defense technologies on satellite networks, it is particularly necessary to study their impact on network performance. Since it is too costly to study satellite networks by physical experiments, a mathematical approach to modeling and evaluating their performance is a proven research method. In this paper, a new scheme is provided for evaluating the performance of satellite network moving target defense system using SPN. Based on the advantages of SPN's powerful mathematical model simulation and graphical modeling, the network model is established visually and intuitively, and some instructive conclusions are drawn through theoretical reasoning and experimental analysis. Nevertheless, there are still some issues and limitations that need to be considered and understood:

1.  MTD technology has a more complex and extensive defense scheme, and the scheme studied in the paper based only on the change of the network attack surface, which is still simple and general, and cannot fully reflect the impact of the application of moving target defense technology on the network.
2.  When the problems studied and the network structure involved are more complex, modeling with Stochastic Petri Nets is prone to state space explosion, low efficiency, and complex calculation.

In future work, we can further model and analyze the satellite network moving target defense system based on the change of three other attack surfaces (i.e., data, software, platform), and optimize the configuration of defense techniques based on the research results, so as to provide more comprehensive guidance for satellite network system security protection.

## 7. Conclusions

This paper focuses on the performance evaluation of satellite networks with active defense technology. SPN is used to build performance evaluation models of satellite networks. Then, we theoretically inferred and analyzed the proposed SPN models. After that, we conduct extensive simulations on the PIPE platform, and the influence of different parameters of the active defense technique on the performance of the whole satellite network is evaluated.

The conclusions are as follows: the deployment of MTD technology reduces the overall performance of the satellite network. Change frequency is the key factor to the performance loss and security of the entire satellite network. To maintain the balance of performance-security, the trade-off between communication performance and change frequency will be one of the main points of research on active defense technology. Improving synchronization and migration efficiency, link anti-interference, and transmission capacity, as well as satellite processing service capacity can effectively alleviate this problem. The results we get can be

used for further improvement of active defense technologies, as well as for the design and optimization of satellite network moving target defense systems.

**Author Contributions:** All authors contributed to the development and completion of this paper. Conceptualization and project administration, L.S.; Methodology validation, formal analysis, and writing—original draft preparation, S.D.; writing—review and editing, Y.M.; visualization, S.L. All authors have read and agreed to the published version of the manuscript.

**Funding:** This research was funded by the Shandong Provincial Natural Science Foundation under Project ZR2019MF034, in part by the Research Funds from Guangxi Key Laboratory of Cryptography and Information Security under Grant GCIS201811, in part by the National Natural Science Foundation of China (NSFC) under Grant No. 61772551.

**Institutional Review Board Statement:** Not applicable.

**Informed Consent Statement:** Not applicable.

**Data Availability Statement:** All data included in this study are available upon request by contact with the corresponding author.

**Acknowledgments:** The authors thank anonymous reviewers who gave valuable suggestions and useful comments, which will greatly improve the manuscript. The authors would also like to acknowledge the cooperation and invaluable assistance of the whole research team throughout the preparation of the original manuscript.

**Conflicts of Interest:** The authors declare no conflict of interest.

## Abbreviations

The following abbreviations are used in this manuscript:

| | |
|---|---|
| MTD | Moving Target Defense |
| PN | Petri Nets |
| SPN | Stochastic Petri Nets |
| MC | Markov Chain |
| PIPE | Platform Independent Petri Net Editor |
| GEO | Geostationary Earth Orbit |
| EH | End Hopping |
| MSD | Mimic Security Defense |
| TPN | Time Petri Nets |
| CPN | Colored Petri Nets |
| NITRD | Networking and Information Technology Research and Development |
| DTN | Delay/Disruption Tolerant Network |
| CR | Cognitive Radio |
| STCPN | Timed Stochastic Colored Petri Nets |
| ICS | Industrial Control System |
| GSPN | Generalized Stochastic Petri Nets |
| PTPN | P-Timed Petri Nets |
| VUP | Vulnerability, Uncertainty, and Probability |
| LEO | Low Earth Orbit |
| CTMC | Continuous Time Markov Chain |

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
