# Peer review of "Modeling and Performance Analysis of Satellite Network Moving Target Defense System with Petri Nets"

_remotesensing, doi:10.3390/rs13071262_

Round 1

Reviewer 1 Report

The research presented in the article is interesting and valuable. The way of presentation is logical, and step by step presenting ideas of authors. In the introduction part, I would suggest emphasizing the objective of the article and research which is presented. Research methodology presented well, however, the discussion and conclusions could emphasize scientifically new findings and most important aspects. As well the limitation and further research trends should be presented in the conclusions part.

Author Response

Dear Reviewer,

We feel great thanks for your professional review work on our article. As you are concerned, there are several problems that need to be addressed. According to your nice suggestions, we have made some corrections to our previous draft and explained some of the questions you raised. Please see the attachment.

Reviewer 2 Report

This study focuses on the performance evaluation of satellite network moving target defense system. Firstly, two Stochastic Petri Nets (SPN) models are constructed to analyze the performance of satellite network in traditional and active defense states respectively. Secondly, the steady-state probability of each marking in SPN models is obtained by using the isomorphism relation between SPN and Markov Chains (MC), and further key performance indicators such as average time delay, throughput and the utilization of bandwidth are reasoned theoretically. Finally, the proposed two SPN models are simulated based on the PIPE platform. I have the following concerns.

  1. The literature survey is poor with critical analysis. Latest literature should be reviewed.
  2. Research gaps should be clearly presented.
  3. What do you mean by "Simulation Experiments"?
  4. More simulation work should be conducted.
  5. Each modelling should be supported by a reference. 
  6. I will recommend summarising the literature as a table.
  7. I can not see Figure 7 in the .pdf file.
  8. A discussion section indicating the advantage and disadvantage of the method should be included.
  9. I will recommend comparing the proposed performance evaluation scheme with existing ones.
  10. Writing quality could be improved.

Author Response

(The authors gave the same response as above.)

Reviewer 3 Report

The paper is fine and the contribution is clear. Nonetheless, I suggest the following changes:

1)The paper is too verbose. Almost all of the paragraphs in the introduction are repeated in some other sections. Please remove or shorten repeated parts.

2)Please create a section "Background". Authors can include in this section "PetriNets concepts" and "active defense techniques" (not only MTD)  which is explained in another section. This section should be before the "related work" section.

3) Please include a short review of research works about  "active defense in satellite networks"  in the "related work" section.

4) Please provide an acronyms list (MDPI's template has such a section). There are too many acronyms without meaning.

Author Response

(The authors gave the same response as above.)

Round 2

Reviewer 2 Report

The quality of the manuscript has been improved.